# Morphology of Dried Drop Patterns of Saliva from a Healthy Individual Depending on the Dynamics of Its Surface Tension

**Lyudmila V. Bel'skaya** [1,*] **, Elena A. Sarf** [1] **and Anna P. Solonenko** [2,3]

1    Department of Biology and Biological Education, Omsk State Pedagogical University, 14 Tukhachevsky str, Omsk 644043, Russia; nemcha@mail.ru
2    Faculty of Dentistry, Omsk State Medical University, 12 Lenina str, Omsk 644099, Russia; anna.petrovna@bk.ru
3    Faculty of Radio Engineering, Nanoengineering, Omsk State Technical University, 11 Mira str., Omsk 644050, Russia
*    Correspondence: Ludab2005@mail.ru; Tel.: +7-913-641-35-77

**Abstract:** *Background*: The study of processes in the drying drops of biological fluids (dried drop patterns) and the method of dynamic surface tensiometry can be attributed to integral methods of assessing the state of the organism. *Research objective*: to establish the relationship between the type of crystallization patterns and the surface tension of human saliva in normal conditions. *Methods*: 100 volunteers (40 males, 60 females) that were aged 30–59 participated in the study. In all saliva samples, the parameters of dynamic tensiometry, types of crystallization patterns and 11 biochemical parameters were determined. *Results*: No statistically significant differences in the saliva crystallization patterns were observed, depending on the age and gender characteristics of the volunteers. A negative correlation of the area of the crystallization zone and the surface tension of saliva is shown. When considering the crystallization patterns, their considerable variability was noted; on this basis, the entire studied sample was divided into four clusters by surface tension. *Conclusion*: In general, the crystallization patterns that are inside the selected groups remain quite heterogeneous. This increases the likelihood of making an incorrect diagnosis when using visual methods to evaluate the crystallization patterns, which significantly limits the use of such diagnostic methods in clinical practice.

**Keywords:** dried drop; crystallization patterns; dynamic surface tension; saliva; age and gender features; biochemical composition; correlations

---

## 1. Introduction

The study of processes in the drying drops of biological fluids (dried drop patterns) and the method of dynamic surface tensiometry can be attributed to integral methods of assessing the state of the organism [1–4]. Thus, during the dehydration of biological fluids, the structure formation of the solid phase proceeds while taking into account the interrelations between the individual parameters of the medium, which allows for them to be analyzed. In this connection, it is possible to use the features of crystallization patterns of biological body fluids with the purpose of identifying various pathological processes [5–8]. In particular, human oral fluid (saliva) can be promising for this purpose [9–12]. The use of saliva in the clinical laboratory diagnosis is a promising, and non-invasive, simple, and inexpensive method for the detection of diseases [13,14]. The integration of complex techniques, including genomic research, epigenome, transcriptome, proteome, metabolome and microbiome, can detect and quantify the number of biomarkers in saliva [15–18]. The study of saliva refers to non-invasive methods and it is

conducted to assess the age and physiological status, identify somatic diseases, pathology of the salivary glands and tissues of the oral cavity, genetic markers, monitoring of drugs, etc. [19–21]. However, it should be noted that, today, there is not a sufficiently convincing theory to describe the processes occurring during the dehydration of body fluids [22–24]. It is not known exactly which factors lead to the formation of certain crystal structures [25–27]. Diagnostic methods that were developed on the basis of the crystallization of biological fluids use a comparison of the crystallization patterns of a healthy and sick person. Therefore, it is necessary to accumulate a sufficiently large array of experimental data for each disease before it can be compared with a high degree of reliability. The difficulties of using this method are also related to the fact that the crystallization pattern of biological fluids, even normally, can vary significantly. This is due to the variability of the biochemical composition of biological fluids, which determines the change in surface tension. One of the limitations in the widespread use of saliva is the high variability of its normal composition, which requires special attention towards the procedure of sample collection and the establishment of reference limits for the content of individual components [28–31]. In turn, the method of dynamic surface tensiometry can also be used to detect changes in the biological fluids on the background of various diseases [32–35]. It is important to note that the dynamic surface tension of biologic liquids is essentially more sensitive to various pathologies than the corresponding equilibrium surface tension values [36,37]. Therefore, dynamic surface tensiograms can be regarded as a comprehensive indicator of some pathologic disturbances, and they have large potential for differential diagnosis and monitoring of the efficiency of therapies. However, its limitations practically coincide with those that are listed above for the crystallization of biological fluids. It is interesting to compare the data regarding the crystallization of biological fluids, its biochemical composition, and the nature of the change in surface tension, which has not yet been carried out for saliva.

The aim of the study was to establish the relationship between the type of crystallization patterns and the surface tension of the saliva of healthy volunteers.

## 2. Materials and Methods

### 2.1. Participants

The saliva samples were obtained from healthy adult male (n = 40) and female (n = 60) volunteers that were aged 30–59 years. All of the volunteers had to be free of fever and/or cold; non-smokers; and, have good oral hygiene, while participants with gingival and periodontal inflammation were excluded. To minimize any contamination of samples and to obtain a relatively constant baseline, the participants were asked to refrain from brushing their teeth and eating or drinking in the 60 min. prior to sample collection. The study was carried out in accordance with the Helsinki Declaration (adopted in June 1964 in Helsinki, Finland and revised in October 2000 in Edinburgh, Scotland) and was approved at a meeting of the Ethics Committee of the Omsk Regional Clinical Hospital "Clinical Oncology Center" on July 21, 2016 (Protocol No. 15). All of the volunteers provided written informed consent.

### 2.2. Collection, Processing and Storage of Saliva Samples

Unstimulated whole expectorated saliva (5 mL) was collected from each subject between 8 and 10 a.m., while considering the circadian rhythm [38]. Subjects rinsed their mouth with water 10 min. prior to sampling. The unstimulated whole saliva samples were centrifuged ($10,000\times g$ for 10 min) to remove the cellular debris and to minimize the turbidity of saliva, which could negatively impact on the accuracy of analysis [39]. The supernatant from each volunteer was divided into eleven aliquots. The biochemical parameters were immediately analyzed after centrifugation (without freezing).

## 2.3. Crystallization of Saliva Samples

The important role of surface wettability in the formation of drying droplets of saliva is known [2]. Such a drying patterns difference was attributed to the fact that the evaporation flux from the peripheral region of a low contact angle drop on a substrate ($\theta < 40°$) is higher than that from the central part of the drop. However, for a saliva drop that was deposited on a non-wetting surface ($\theta > 90°$), the difference in the evaporation flux of the peripheral region and the central part of the drop would be small [40]. Therefore, the drying of saliva drop on the non-wetting substrates would yield different patterns from those on the wetting substrates [41]. In this study, a skimmed glass was used to obtain dried droplets of saliva, because it has better wettability.

The dosing of the saliva was carried out in the micro-burette of total volume 10 μL on a clean slide. It is suitable to dose three droplets on one slide for comparison and reproducibility. The samples were crystallized at room temperature (about 25 °C) and with a relative air humidity of 60% in a dust free environment. The evaluation of the created samples of crystals was carried out using a light microscope (magnification 40).

## 2.4. Surface Tension Measurements

For the measurements of dynamic surface tension γ in the relevant short lifetime range, the maximum bubble pressure method (MBPM) is the most appropriate for medical applications involving biological liquids [42,43]. The main advantages of MBPM are the small sample volume (1 mL) and the wide range of surface lifetime (0.001 s to 30–50 s).

Saliva surface tension was determined using the DSA30S system (KRUSS, Hamburg, Germany), which was controlled via the ADVANCE software (KRUSS). The measurements were carried out using the hanging drop method that is based on the analysis of the shape of a saliva drop formed on the point of a needle in air medium (t = 20 °C). Using the automated dosage system, a stably hanging saliva drop about 5 microliters in volume was formed on a point of a stainless steel needle 1.83 micron in diameter. Further, the measurement program was initiated, during which the σ of saliva was calculated using the ADVANCE software by the Jung-Laplace equation, according to the geometric parameters of the drop determined each second for 5 min. A plot of saliva σ vs. measurement time was formed according to the obtained values. The most informative surface tension values are $\gamma_{0.01}$, $\gamma_{1.0}$, $\gamma_{max}$, and $\gamma_{\infty}$, i.e., values obtained at times $t = 0.01$ s, $t = 1$ s, $t = 100$ s and extrapolated to $t \to \infty$, respectively. Additionally, the slope values of tensiometric curve plotted in the coordinates $\gamma(t^{1/2})$ and $\gamma(t^{-1/2})$ were calculated as $\lambda_0 = -(d\gamma/dt^{1/2})_{t \to 0}$ and $\lambda_{\infty} = (d\gamma/dt^{-1/2})_{t \to \infty}$, respectively.

## 2.5. Biochemical Analysis of Saliva

In all of the saliva samples, 11 biochemical parameters were determined, including pH, mineral, and protein contents. The total calcium content (mmol/L) was photometrically determined using the semi-automatic StatFax 3300 biochemical analyzer according to a reaction with Arsenazo III, magnesium was determined by reaction with xylidyl blue, phosphorus was determined by reaction with ammonium molybdate, chlorides were determined by a reaction with mercury thiocyanate using the kits from Vektor-Best LLC (Novosibirsk). The concentration of urea (mmol/L) was determined by the photometric urease and salicylate method, according to Berthelot, that of total protein (g/L) was determined by reaction with pyrogallol red, and that of albumin (g/L) was determined by reaction with bromocresol green. The KAPEL-105M capillary electrophoresis system was used to determine the concentration of potassium and sodium ions (Lumex, St. Petersburg, Russia) [44]. The level of seromucoids was determined using the turbidimetric method, according to Huergo.

The pH meter was accurate to ±0.002 pH units and a three point calibration was used at pH 4, 7, and 10. The pH of human saliva was immediately measured after collection in triplicate for each participant. Photometric methods were used to determine the concentrations of calcium, phosphorus, magnesium, chlorides, protein, albumin, urea, and seromucoids. A standard calibration method was

used for measurement and calculation (Standard Mode). The resolution of the analyzer allows for you to fix the optical density of ±0.0002 units, the accuracy of analyte concentration is determined by the concentration of calibrators (calcium −2.50 mmol/L, phosphorus −1.61 mmol/L, magnesium −0.328 mmol/L, chlorides −100 mmol/l, protein −0.50 mg/L, albumin −40 g/L, urea −8.33 mmol/L). The error of the method does not exceed 1–2%. Seromucoids concentration is expressed in units of optical density. The concentration of potassium and sodium was calculated according to a previously constructed calibration curve and the error of the method does not exceed 10%. All of the measurements were carried out in triplicate to ensure reproducible results.

### 2.6. Statistical Methods

The statistical analysis of the obtained data was performed by means of Statistica 10.0 (StatSoft) program and R package (version 3.2.3) while using the non-parametric method and Wilcoxon criterion in the dependent groups and Mann–Whitney U-criterion in the independent groups. Preliminary emissions check has been executed using the Grubbs test. The sample was described by calculating the median (Me) and interquartile range in the form of the 25th and 75th percentile [LQ; UQ]. The differences were considered to be statistically significant at $p < 0.05$. The statistical interrelations were studied using the nonparametric correlation analysis by performing the calculation of Spearman correlation coefficients (R).

## 3. Results

### 3.1. Zones in the Dried Drop Patterns of Saliva

It is known that the dehydration of a drop of protein-salt solutions, including saliva, during drying leads to a gradual increase in the concentration of salts, which stimulates the process of protein phase separation and the formation of structures in protein ash [45–48]. The dendritic structures in dried drops of protein-salt solutions have a salt nature. The optically empty zone is visible around each large crystal, which results from the depletion of salt solution, retreating from the center of the crystal to the periphery due to evaporation of water. The protein layers between salt patterns remain dark. Thus, the salt and protein structures in the dried drops are clearly distinguishable.

In the dried drop patterns of saliva, a clear systemic structure was observed with division into two zones: peripheral amorphous and central crystalline. Three types of crystallization patterns were established, which differed from each other in salt crystals and amorphous structures. The first type had the largest salt crystal area of 70–85% (Figure 1a), the second type was characterized by a crystallization zone of 30–35% (Figure 1b), and the third with-8–10% (Figure 1c).

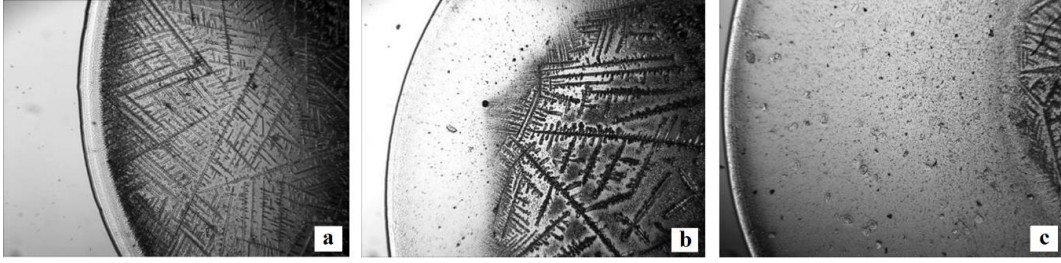

**Figure 1.** Zones in the dried drop patterns of saliva (×40).

It is shown that the average value of the area that is occupied by salt crystals ($S_{cr}$) was 83.38%, which is due to the predominance of the first type of crystallization. In this case, the second type of crystallization was encountered 17 times in the sample under study, while the third type - 1 time.

It should be noted that, for samples characterized by the maximum area of the crystallization zone, the surface tension has the lowest values (Table 1). Moreover, it concerns the surface tension at the initial moment of time, which reflects the influence of surface-active substances in high concentrations.

**Table 1.** Parameters of saliva samples depending on area of crystallization zone.

| Parameters | 1 Type of Crystallization n = 82 | 2 Type of Crystallization n = 17 | 3 Type of Crystallization n = 1 |
|---|---|---|---|
| $S_{cr}$, % | 88.9 [78.2; 92.9] | 54.9 [48.9; 60.9] | 12.4 |
| | - | *p = 0.0000* | - |
| **Tensiometric parameters** | | | |
| $\gamma_{0.01}$, mN/m | 63.19 [60.01; 68.47] | 65.61 [62.74; 67.44] | 73.56 |
| $\gamma_{1.0}$, mN/m | 60.26 [56.27; 64.26] | 62.42 [57.90; 65.39] | 71.77 |
| $\gamma_{max}$, mN/m | 54.78 [49.12; 58.23] | 54.49 [50.77; 58.27] | 65.47 |
| $\gamma_{\infty}$, mN/m | 46.35 [35.22; 50.34] | 42.48 [34.95; 47.39] | 53.83 |
| **Biochemical parameters** | | | |
| pH | 6.61 [6.47; 6.77] | 6.79 [6.51; 6.99] | 6.88 |
| Calcium, mmol/L | 1.11 [0.77; 1.47] | 0.96 [0.75; 1.48] | 0.21 |
| Phosphorus, mmol/L | 4.70 [3.95; 5.61] | 4.32 [4.05; 4.89] | 3.20 |
| Ca/P | 0.237 [0.196; 0.262] | 0.221 [0.185; 0.303] | 0.066 |
| Sodium, mmol/L | 4.2 [2.9; 5.9] | 3.8 [2.4; 6.5] | 5.4 |
| Potassium, mmol/L | 10.2 [7.4; 13.4] | 9.2 [6.3; 12.1] | 8.0 |
| Na/K | 0.42 [0.39; 0.44] | 0.42 [0.39; 0.53] | 0.68 |
| Chlorides, mmol/L | 18.5 [14.6; 24.3] | 13.8 [13.3; 19.8] | 6.61 |
| | - | *p = 0.0141* | - |
| Magnesium, mmol/L | 0.229 [0.161; 0.325] | 0.226 [0.195; 0.331] | 0.051 |
| Protein, g/L | 0.72 [0.54; 0.84] | 0.61 [0.45; 0.82] | 0.10 |
| Urea, mmol/L | 7.84 [6.36; 9.59] | 5.70 [4.59; 7.86] | 1.55 |
| | - | *p = 0.0332* | - |
| Albumin, g/L | 0.28 [0.20; 0.41] | 0.24 [0.12; 0.38] | 0.04 |

Note. The ratios Ca/P and Na/K characterize the processes of mineralization and demineralization in the oral cavity [38].

However, the total content of surface-active substances in the group with type 1 crystallization of saliva is higher, which is consistent with a higher content of protein substances. For the same group, the maximum content of mineral substances in saliva was noted. This fact can serve as a confirmation of the theory that the first nuclei of crystals appears in the peripheral zone due to the local supersaturation of the solution in the boundary region. A large concentration of mineral substances in the samples promotes more rapid achievement of the supersaturation of the solution, which is necessary for the onset of crystallization, whereas the third type of crystallization requires the evaporation of almost the entire drop for the appearance of crystals (Figure 1). Later, crystallization can proceed in two ways: the deposition of a solid phase on the surface of already existing crystals with the mutual penetration of some growing dendritic crystals into other dendrites (Figure 2a) or the formation of new crystalline nuclei (Figure 2b).

The predominance of one of these types of growth could be caused by the composition of the biological fluid and the action of surface forces. Thus, it is more energetically advantageous for ionic particles to join the tops and kinks of a growing crystal, which, if there are dislocations on the crystal,

can cause the formation of dendrites (Figure 2a). The presence of impurities that could be adsorbed by the growing surface of a crystal could specifically change the growth of faces, as well as hinder the growth of already existing crystals (Figure 2b).

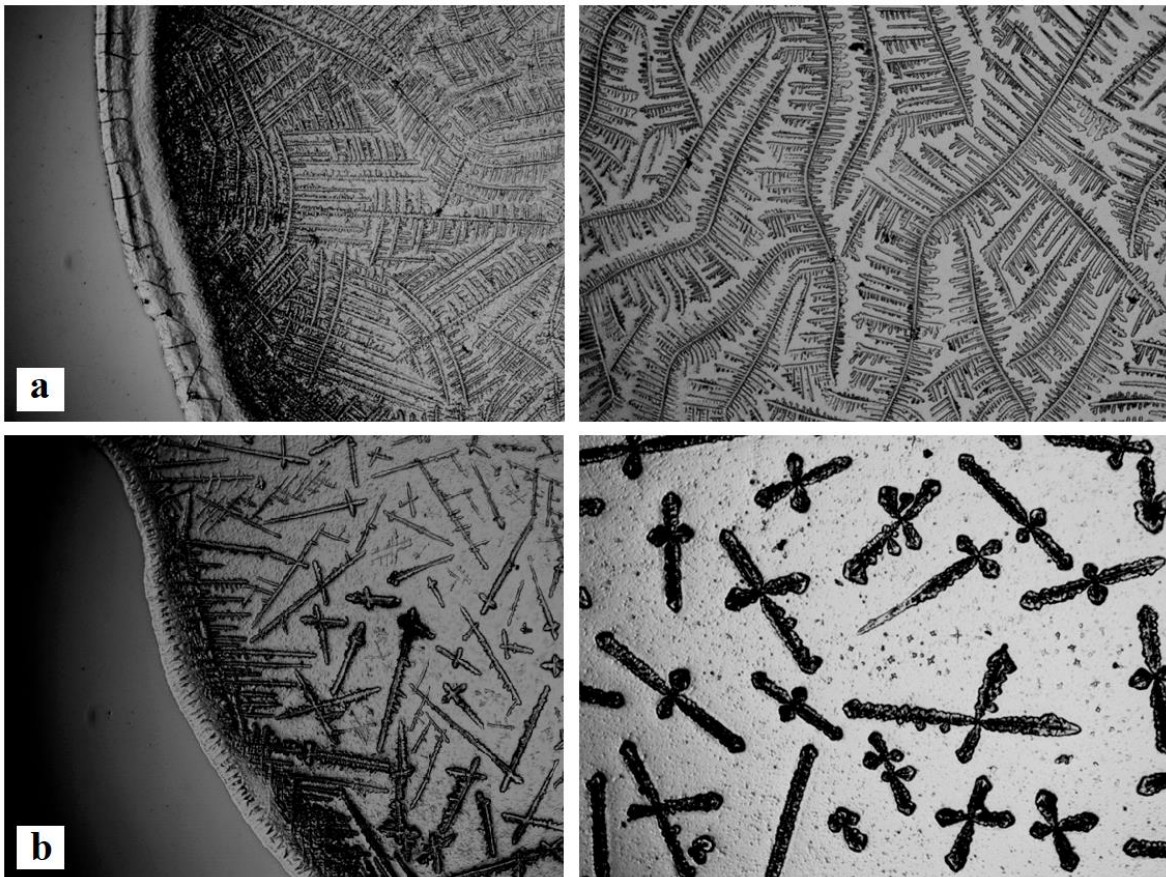

**Figure 2.** The morphological features of dried drop patterns of saliva: (**a**)—dendritic growth, (**b**)—new crystal nuclei (on the left—the peripheral zone, on the right—the central zone) (×40).

### 3.2. The Influence of Gender and Age Characteristics on Morphological Features of Dried Drop Patterns of Saliva

At the next step of the study, the determination of morphological features of dried drop patterns of saliva was carried out depending on the gender (Table 2).

**Table 2.** Parameters of saliva samples depending on gender.

| Parameters | Females n = 60 | Males n = 40 | *p*-Value |
|---|---|---|---|
| $S_{cr}$, % | 84.3 [72.3; 92.0] | 82.3 [65.3; 93.5] | 0.8922 |
| $\gamma_{0.01}$, mN/m | 62.50 [59.78; 67.44] | 65.66 [61.13; 69.46] | 0.0883 |
| $\gamma_{1.0}$, mN/m | 59.95 [56.25; 63.42] | 63.14 [58.07; 66.75] | 0.0271[*] |
| $\gamma_{max}$, mN/m | 54.08 [50.06; 57.62] | 55.68 [48.58; 60.09] | 0.1895 |
| $\gamma_{\infty}$, mN/m | 45.46 [34.86; 49.86] | 46.37 [35.06; 52.30] | 0.5879 |
| $\lambda_0$, mN·m$^{-1}$·s$^{-1/2}$ | 1.65 [1.29; 3.19] | 1.89 [1.41; 2.97] | 0.4817 |
| $\lambda_{\infty}$, mN·m$^{-1}$·s$^{-1/2}$ | 0.48 [0.39; 0.64] | 0.56 [0.30; 0.76] | 0.3407 |

Note. * - the differences are statistically significant ($p < 0.05$).

The studied group was additionally split into subgroups according to age with 10 years range: 30–39, 40–49, and 50–59 (Table 3) [49].

**Table 3.** Parameters of saliva samples depending on gender and age.

| Parameters | 30–39 Years (1) | 40–49 Years (2) | 50–59 Years (3) |
|---|---|---|---|
| | | **Males** | |
| Group size | n = 8 | n = 12 | n = 20 |
| Age, years | 33.0 [28.6; 37.4] | 44.6 [42.0; 47.2] | 53.6 [50.9; 56.7] |
| | - | $p_{1-2} < 0.0001$ | $p_{1-3} < 0.0001$; $p_{2-3} < 0.0001$ |
| $S_{cr}$, % | 74.8 [65.2; 91.1] | 83.8 [75.8; 91.6] | 86.8 [77.2; 92.6] |
| $\gamma_{0.01}$, mN/m | 66.77 [64.03; 70.17] | 65.61 [62.77; 69.49] | 65.64 [60.23; 68.71] |
| $\gamma_{1.0}$, mN/m | 63.81 [57.83; 67.42] | 64.10 [57.98; 69.11] | 61.85 [58.39; 65.28] |
| $\gamma_{max}$, mN/m | 56.63 [48.52; 60.70] | 56.43 [47.28; 60.06] | 55.68 [51.00; 59.60] |
| $\gamma_{\infty}$, mN/m | 45.28 [35.47; 53.75] | 44.95 [30.28; 52.46] | 46.37 [35.19; 50.32] |
| $\lambda_0$, mN·m$^{-1}$·s$^{-1/2}$ | 1.97 [1.49; 3.55] | 1.75 [1.27; 4.32] | 1.89 [1.50; 2.65] |
| $\lambda_{\infty}$, mN·m$^{-1}$·s$^{-1/2}$ | 0.54 [0.32; 0.76] | 0.41 [0.24; 0.58] | 0.66 [0.48; 0.85] |
| | - | - | $p_{2-3} = 0.0027$ |
| | | **Females** | |
| Group size | n = 10 | n = 20 | n = 30 |
| Age, years | 34.9 [30.6; 38.5] | 46.8 [43.4; 48.6] | 54.3 [52.2; 57.5] |
| | - | $p_{1-2} < 0.0001$ | $p_{1-3} < 0.0001$; $p_{2-3} < 0.0001$ |
| $S_{cr}$, % | 80.8 [62.1; 91.8] | 81.8 [71.9; 94.7] | 78. 6 [64.9; 94.2] |
| $\gamma_{0.01}$, mN/m | 67.73 [62.74; 70.43] | 61.38 [56.86; 63.97] | 61.86 [60.26; 65.62] |
| $\gamma_{1.0}$, mN/m | 65.31 [58.95; 67.16] | 58.31 [54.28; 61.49] | 59.47 [56.27; 62.30] |
| | - | $p_{1-2} = 0.0122$ | $p_{1-3} = 0.0244$ |
| $\gamma_{max}$, mN/m | 55.44 [53.49; 60.40] | 52.90 [48.19; 56.50] | 53.26 [50.37; 57.62] |
| $\gamma_{\infty}$, mN/m | 44.04 [34.86; 50.93] | 43.76 [34.87; 49.68] | 45.46 [36.82; 49.20] |
| $\lambda_0$, mN·m$^{-1}$·s$^{-1/2}$ | 3.05 [1.54; 3.84] | 1.51 [1.26; 2.14] | 1.56 [1.29; 2.91] |
| | - | - | $p_{1-3} = 0.0049$ |
| $\lambda_{\infty}$, mN·m$^{-1}$·s$^{-1/2}$ | 0.46 [0.39; 0.64] | 0.47 [0.31; 0.66] | 0.49 [0.40; 0.61] |

There were no significant differences in the nature of the crystallization of saliva, depending on age and gender characteristics. It was shown that saliva surface tension in male samples is higher than in females; however, statistically reliable differences were only found for $\gamma_{1.0}$ (Table 2). At the isolation of three age groups in the sample, the same trend is maintained: surface tension in female samples is slightly lower than in male samples in each age group, with an exception in the 30–39 age group, for which at small lifetimes of the drop surface the surface tension is higher in the female group (Table 3). It should be noted that, in general, at short times, surface tension decreases with age, whereas the equilibrial $\gamma_{\infty}$ insignificantly lowers at the transition from the younger age group to the middle one (−0.73% and −0.64%), and then increases (+3.2% and +3.9% for males and female samples, correspondingly).

When comparing the inclination angles in dynamic tensiograms in the $\gamma(t^{-1/2})$ coordinates, information regarding the total concentration of surfactant components in the studied biological fluid sample could be obtained. As it is seen from the outlined data (Table 3), no statistically reliable differences were found in the values of inclination angles and the contents of the surfactants between the studied groups. However, it was established that, when simultaneously taking into the account of age and gender of the studied persons (Table 2), the surfactant content is maximum for the younger age group (30–39), both for females and for males. Subsequently, it decreases at transition to the middle

group (−11.2% and −50.5% for males and females correspondingly) and further it insignificantly increases (+8.0% for males and +3.3% for females).

Biochemical analysis of saliva in selected groups showed an increase in the protein content with age, while increasing the concentration of mineral substances for both males and females (Table 4).

**Table 4.** Biochemical parameters of saliva samples depending on gender and age.

| Parameters | 30–39 Years (1) | 40–49 Years (2) | 50–59 Years (3) |
|---|---|---|---|
| **Males** | | | |
| pH | 6.62 [6.43; 6.76] | 6.84 [6.52; 7.00] | 6.73 [6.52; 6.85] |
| Calcium, mmol/L | 1.05 [0.75; 1.54] | 1.16 [0.84; 1.54] | 1.13 [0.76; 1.55] |
| Phosphorus, mmol/L | 4.77 [4.42; 5.92] | 5.52 [4.71; 6.14] | 4.52 [3.77; 5.63] |
| Sodium, mmol/L | 2.9 [2.2; 4.2] | 4.4 [2.7; 5.7] | 5.8 [4.1; 8.8] |
| | - | - | $p_{1-3} = 0.0168$ |
| Potassium, mmol/L | 7.7 [6.8; 10.7] | 9.4 [8.0; 14.4] | 11.1 [8.3; 13.5] |
| | - | - | $p_{1-3} = 0.0433$ |
| Chlorides, mmol/L | 15.9 [15.1; 19.9] | 18.3 [14.6; 21.9] | 19.2 [16.2; 23.6] |
| Magnesium, mmol/L | 0.194 [0.135; 0.288] | 0.180 [0.143; 0.281] | 0.259 [0.198; 0.374] |
| Protein, g/L | 0.66 [0.47; 0.83] | 0.80 [0.68; 0.91] | 0.78 [0.61; 0.90] |
| Urea, mmol/L | 6.96 [6.39; 7.06] | 8.52 [7.25; 9.88] | 9.52 [8.16; 10.99] |
| | - | $p_{1-2} = 0.0206$ | $p_{1-3} = 0.0110$ |
| Albumin, g/L | 0.29 [0.23; 0.36] | 0.22 [0.18; 0.44] | 0.29 [0.18; 0.44] |
| **Females** | | | |
| pH | 6.67 [6.45; 6.98] | 6.60 [6.51; 6.76] | 6.61 [6.47; 6.71] |
| Calcium, mmol/L | 0.88 [0.68; 1.39] | 1.04 [0.59; 1.24] | 1.20 [0.65; 1.48] |
| Phosphorus, mmol/L | 3.39 [2.67; 4.22] | 4.66 [3.31; 5.63] | 4.41 [4.13; 5.02] |
| | - | - | $p_{1-3} = 0.0244$ |
| Sodium, mmol/L | 3.9 [2.6; 8.1] | 3.5 [2.7; 7.6] | 4.1 [2.5; 5.5] |
| Potassium, mmol/L | 8.5 [5.3; 9.4] | 10.4 [6.2; 13.1] | 11.5 [6.6; 14.3] |
| Chlorides, mmol/L | 13.7 [13.0; 21.6] | 17.1 [12.6; 21.4] | 21.6 [15.1; 26.7] |
| | - | - | $p_{1-3} = 0.0478$ |
| Magnesium, mmol/L | 0.267 [0.125; 0.351] | 0.193 [0.068; 0.258] | 0.277 [0.226; 0.328] |
| Protein, g/L | 0.47 [0.39; 0.54] | 0.62 [0.43; 0.83] | 0.68 [0.55; 0.84] |
| | - | - | $p_{1-3} = 0.0054$ |
| Urea, mmol/L | 5.45 [4.05; 7.19] | 6.80 [6.10; 8.53] | 7.38 [6.04; 9.45] |
| | - | - | $p_{1-3} = 0.0498$ |
| Albumin, g/L | 0.20 [0.12; 0.26] | 0.30 [0.18; 0.44] | 0.30 [0.23; 0.47] |
| | - | $p_{1-2} = 0.0386$ | $p_{1-3} = 0.0110$ |

The combination of these two factors, as shown above, leads to an increase in the area of the crystallization zone for the saliva samples of people of the older age group (Table 3).

### 3.3. The Relationship of Surface Tension and Morphological Features of Dried Drop Patterns of Saliva

It should be noted that, at an examination of morphological features of dried drop patterns of saliva, their significant variability was noted, even within the limits of the isolated groups; therefore, at the following step of the study, the entire studied sample was divided into clusters according to surface tension using cluster analysis (Statistica, StatSoft). The separation into four clusters (Figure 3) characterized by different surface tension dynamics over time is shown (Figure 4).

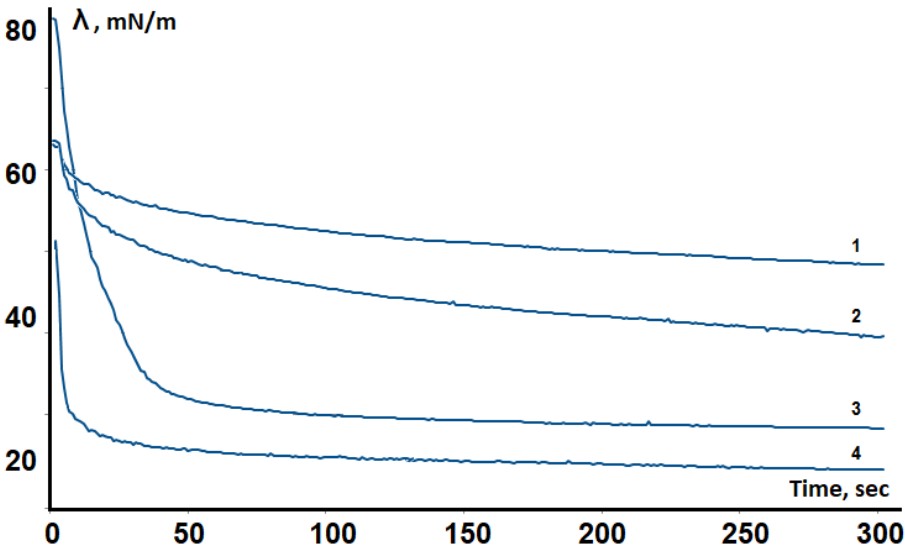

**Figure 3.** Examples of saliva tensiograms for isolated clusters.

The parameters of the dynamic interphase tensiometry of the saliva were calculated for each cluster (Table 5).

**Table 5.** Parameters of saliva samples depending on the cluster.

| Parameters | Cluster 1, n = 39 (1) | Cluster 2, n = 37 (2) | Cluster 3, n = 8 (3) | Cluster 4, n = 15 (4) |
|---|---|---|---|---|
| $S_{cr}$, % | 86.8 [71.8; 92.0] | 81.0 [62.3; 92.0] | 90.9 [80.8; 95.3] | 81.8 [71.5; 92.6] |
| $\gamma_{0.01}$, mN/m | 65.64 [63.19; 69.69] | 60.64 [58.81; 62.76] | 83.59 [75.07; 89.00] | 56.95 [49.61; 67.89] |
| | - | $p_{1\text{-}2} < 0.0001$ | $p_{1\text{-}3} = 0.0001$; $p_{2\text{-}3} < 0.0001$ | $p_{1\text{-}4} = 0.0342$; $p_{3\text{-}4} = 0.0033$ |
| $\gamma_{1.0}$, mN/m | 64.05 [61.70; 66.62] | 58.61 [56.59; 60.69] | 74.02 [71.95; 75.48] | 47.07 [41.02; 54.92] |
| | - | $p_{1\text{-}2} < 0.0001$ | $p_{1\text{-}3} = 0.0013$; $p_{2\text{-}3} = 0.0004$ | $p_{1\text{-}4} < 0.0001$; $p_{2\text{-}4} < 0.0001$; $p_{3\text{-}4} = 0.0002$ |
| $\gamma_{max}$, mN/m | 59.16 [57.26; 61.30] | 52.71 [50.66; 54.49] | 53.65 [47.74; 59.56] | 36.16 [29.79; 41.99] |
| | - | $p_{1\text{-}2} < 0.0001$ | $p_{1\text{-}3} = 0.0203$ | $p_{1\text{-}4} < 0.0001$; $p_{2\text{-}4} < 0.0001$; $p_{3\text{-}4} = 0.0001$ |
| $\gamma_{\infty}$, mN/m | 51.69 [49.52; 53.88] | 43.51 [37.88; 46.59] | 32.89 [31.02; 37.35] | 28.69 [27.08; 30.49] |
| | - | $p_{1\text{-}2} < 0.0001$ | $p_{1\text{-}3} < 0.0001$; $p_{2\text{-}3} = 0.0007$ | $p_{1\text{-}4} < 0.0001$; $p_{2\text{-}4} < 0.0001$; $p_{3\text{-}4} = 0.0033$ |
| $\lambda_0$, mN·m$^{-1}$·s$^{-1/2}$ | 1.33 [1.20; 1.88] | 1.75 [1.40; 2.78] | 6.02 [5.22; 7.40] | 3.79 [2.64; 5.40] |
| | - | $p_{1\text{-}2} = 0.0060$ | $p_{1\text{-}3} < 0.0001$; $p_{2\text{-}3} < 0.0001$ | $p_{1\text{-}4} < 0.0001$; $p_{2\text{-}4} < 0.0001$; $p_{3\text{-}4} = 0.0090$ |
| $\lambda_{\infty}$, mN·m$^{-1}$·s$^{-1/2}$ | 0.54 [0.48; 0.68] | 0.56 [0.40; 0.78] | 0.60 [0.23; 0.66] | 0.12 [0.10; 0.20] |
| | - | - | - | $p_{1\text{-}4} < 0.0001$; $p_{2\text{-}4} < 0.0001$; $p_{3\text{-}4} = 0.0017$ |

It is seen that clusters 1 and 2 are maximally close in characteristics; for them, the initially average surface tension is noted, which then gradually insignificantly decreases over time (21.3% and 28.2% for clusters 1 and 2, correspondingly). Cluster 3 is characterized by the maximum of surface tension at the starting moment of time, which quite drastically decreases to values below those for clusters 1 and 2. For this cluster, the difference between the starting and final surface tension is maximal and it is 60.7%. Cluster 4 is characterized by the lowest initial surface tension, which drastically decreases and almost does not change until the final measurement time (−49.6%). It was shown that differences between all of the isolated groups are statistically reliable (Table 5).

It was noted that, for samples that were characterized by the maximum surface tension at the initial moment of time, the area of the crystallization zone is maximal (cluster 3, Figure 4c), whereas for cluster 2 and 4, the opposite is observed (Figures 2b and 4d). The samples of cluster 1 occupy an intermediate position both in the area of the crystallization zone and in $\gamma_{0.01}$ (Figure 4a). As for the equilibrium surface tension $\gamma_{\infty}$, there is no obvious relations between the surface tension and the area of the crystallization zone.

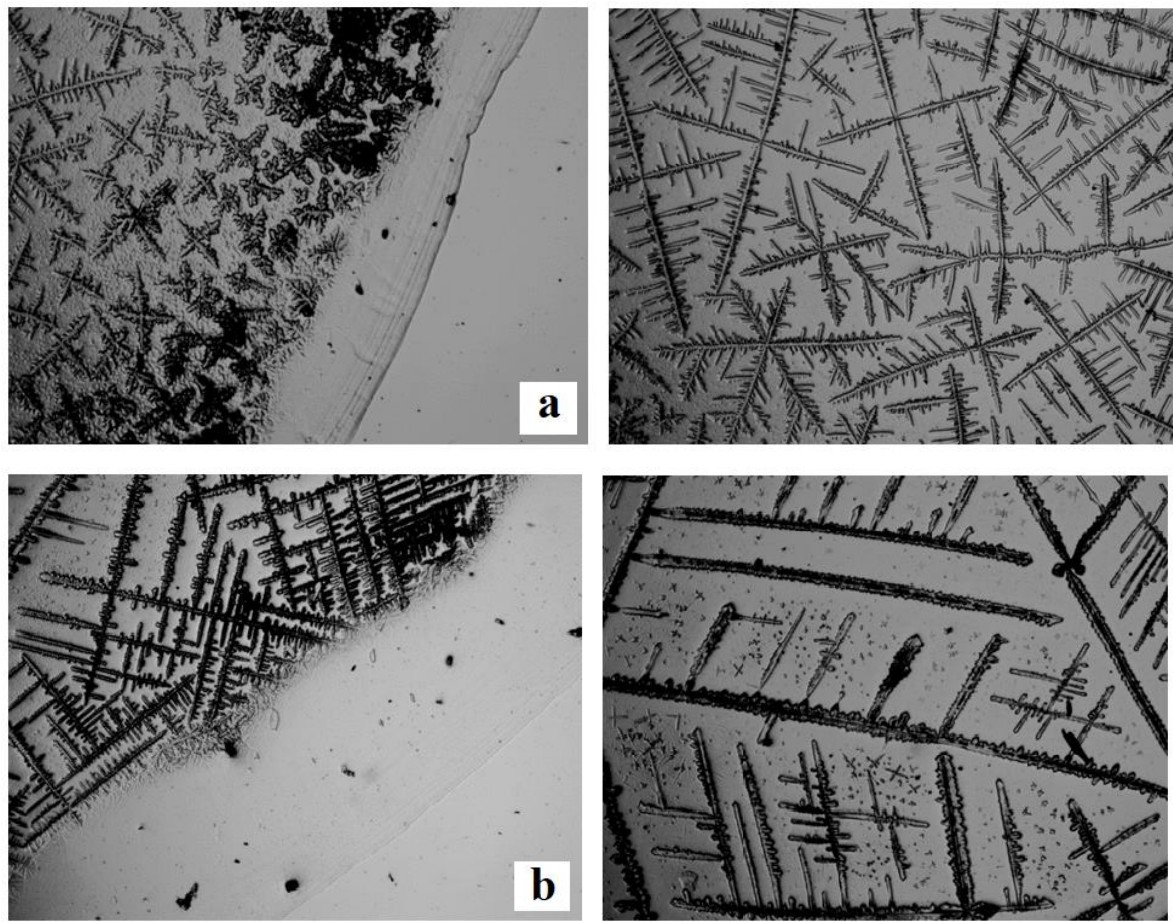

**Figure 4.** *Cont.*

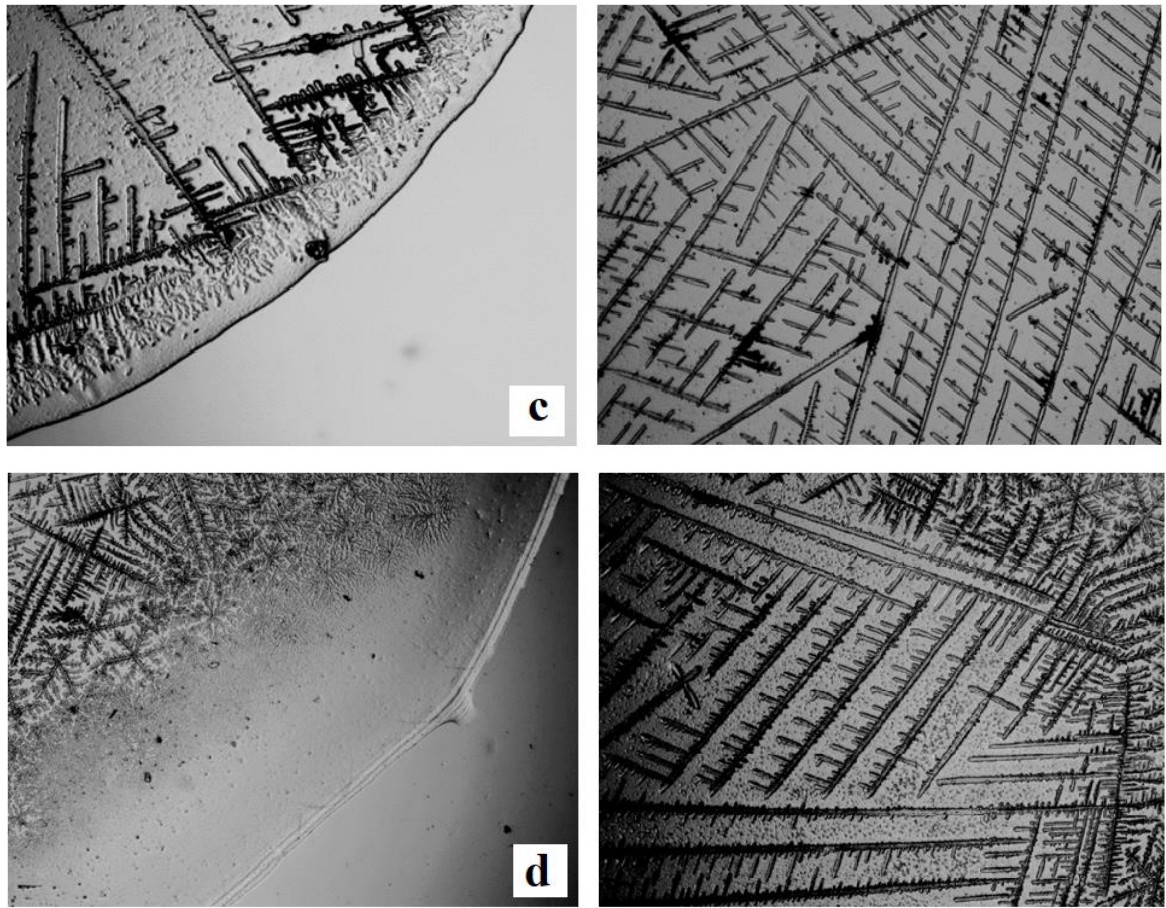

**Figure 4.** The morphological features of dried drop patterns of saliva from (**a**) cluster 1; (**b**) cluster 2; (**c**) cluster 3; and, (**d**) cluster 4 (on the left—the peripheral zone, on the right—the central zone) (×40).

A distinctive feature of cluster 4 is the presence of a pronounced intermediate zone in the crystallization patterns (Figure 5).

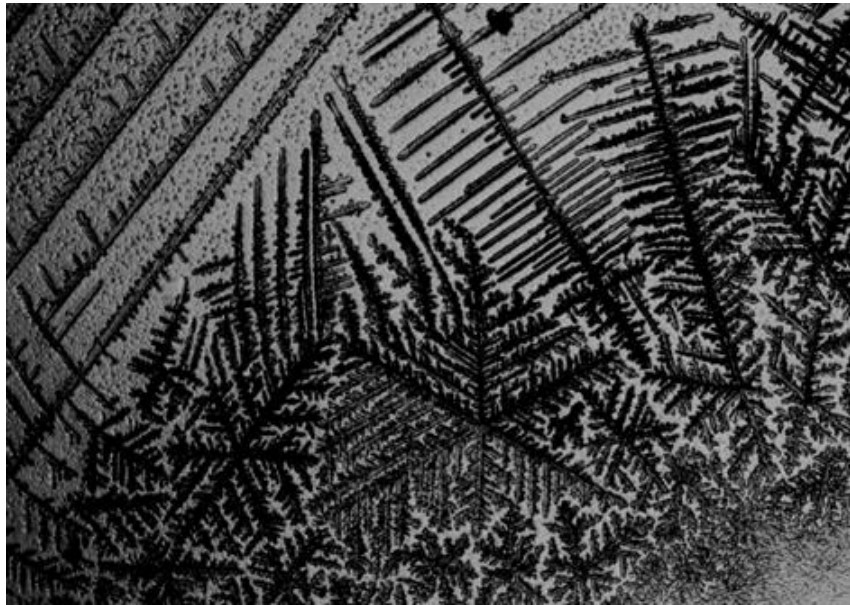

**Figure 5.** Intermediate zone in the crystallization patterns of saliva from cluster 4 (×40).

At the following step of the study, the biochemical saliva content parameters were determined (Table 6). It was established that depending on a certain cluster, the correlation of the biochemical composition, and σ differ. Thus, a positive correlation to pH was found for the cluster 1 *(r = 0.5628; p = 0.0042)* and negative one to the protein and albumin contents *(r = −0.2966; p = 0.0426* and *r = −0.5785; p = 0.0079)*. For the cluster 2, a positive relationship of surface tension with potassium and chloride levels was established *(r = 0.3238; p = 0.0446 and r = 0.5388; p = 0.0092)*. For cluster 3, a negative correlation with the Ca/P ratio was shown *(r = −0.5952; p = 0.0051)*, as well as with protein and albumin levels *(r = −0.7857; p = 0.0002 and r = −0.5270; p = 0.0027)*, and a positive relationship with the Na/K ratio *(r = 0.3571; p = 0.0063)*.

**Table 6.** Biochemical composition of saliva in accordance with isolated clusters.

| Parameters | Cluster 1, n = 39 (1) | Cluster 2, n = 37 (2) | Cluster 3, n = 8 (3) | Cluster 4, n = 15 (4) |
|---|---|---|---|---|
| Age, years | 49.0 [42.4; 52.8] | 51.9 [48.4; 56.8] | 45.0 [41.8; 48.7] | 47.0 [39.8; 53.4] |
| | - | $p_{1-2} = 0.0344$ | $p_{2-3} = 0.0206$ | - |
| pH | 6.58 [6.46; 6.79] | 6.65 [6.45; 6.80] | 6.87 [6.51; 7.04] | 6.72 [6.56; 6.82] |
| | - | - | $p_{1-3} = 0.0578$ | - |
| Calcium, mmol/L | 1.08 [0.68; 1.36] | 1.05 [0.75; 1.48] | 1.13 [0.86; 1.81] | 1.14 [0.73; 1.47] |
| Phosphorus, mmol/L | 4.53 [3.62; 5.65] | 4.41 [3.79; 5.28] | 4.85 [4.11; 5.83] | 5.08 [4.22; 6.60] |
| Ca/P | 0.212 [0.168; 0.296] | 0.238 [0.163; 0.315] | 0.263 [0.176; 0.376] | 0.193 [0.149; 0.284] |
| Sodium, mmol/L | 4.2 [3.3; 5.8] | 4.7 [3.5; 7.4] | 3.3 [2.4; 4.7] | 2.7 [1.6; 4.5] |
| | - | - | - | $p_{1-4} = 0.0083$; $p_{2-4} = 0.0035$ |
| Potassium, mmol/L | 10.3 [7.9; 14.1] | 11.4 [9.2; 14.1] | 4.9 [3.8; 8.8] | 7.4 [6.2; 9.8] |
| | - | - | $p_{1-3} = 0.0059$; $p_{2-3} = 0.0051$ | $p_{1-4} = 0.0232$; $p_{2-4} = 0.0146$ |
| Na/K | 0.38 [0.29; 0.63] | 0.46 [0.32; 0.72] | 0.58 [0.49; 0.96] | 0.36 [0.17; 0.63] |
| Chlorides, mmol/L | 18.3 [15.1; 24.3] | 22.4 [15.3; 24.3] | 14.4 [11.8; 18.5] | 14.3 [13.5; 18.3] |
| | - | - | $p_{1-3} = 0.0559$; $p_{2-3} = 0.0174$ | $p_{1-4} = 0.0332$; $p_{2-4} = 0.0171$ |
| Magnesium, mmol/L | 0.262 [0.196; 0.362] | 0.234 [0.195; 0.311] | 0.131 [0.105; 0.346] | 0.175 [0.091; 0.258] |
| | - | - | - | $p_{1-4} = 0.0031$; $p_{2-4} = 0.0038$ |
| Protein, g/L | 0.73 [0.53; 0.84] | 0.61 [0.46; 0.80] | 0.51 [0.42; 0.77] | 0.79 [0.61; 0.94] |
| | - | - | - | $p_{2-4} = 0.0471$ |
| Urea, mmol/L | 8.47 [6.51; 10.27] | 8.39 [5.70; 10.25] | 6.31 [5.26; 7.71] | 6.65 [5.78; 6.95] |
| | - | - | $p_{1-3} = 0.0617$ | $p_{1-4} = 0.0386$ |

<div align="center">**Table 6.** *Cont.*</div>

| Parameters | Cluster 1, n = 39 (1) | Cluster 2, n = 37 (2) | Cluster 3, n = 8 (3) | Cluster 4, n = 15 (4) |
|---|---|---|---|---|
| Albumin, g/L | 0.30 [0.20; 0.38] | 0.28 [0.22; 0.49] | 0.25 [0.17; 0.37] | 0.19 [0.09; 0.28] |
| | - | - | - | $p_{1-4} = 0.0095;$ $p_{2-4} = 0.0170$ |
| Seromucoids | 0.087 [0.063; 0.143] | 0.098 [0.063; 0.118] | 0.073 [0.028; 0.103] | 0.139 [0.098; 0.169] |
| | - | - | - | $p_{1-4} = 0.0436;$ $p_{2-4} = 0.0348;$ $p_{3-4} = 0.0139$ |

## 4. Discussion

Despite the fact that water content in the saliva varies from 98.5% to 99.5%, even at low lifetimes of the drop surface, the surface tension of the saliva is lower than the corresponding values for redistilled water (72.7 mN/m according to our data), both for male and for female samples. The decrease in surface tension is determined by the biochemical composition of the saliva, which is confirmed by the calculation of correlation coefficients according to Spearman ($p < 0.05$). Thus, a negative correlation of surface tension with protein and albumin contents in the saliva was noted. We have identified a positive correlation of surface tension with the concentration of potassium, sodium, and chloride ions, despite the fact that saliva electrolytes are surface inactive compounds.

It is known that saliva is a structured biological fluid, the entire volume of which is distributed between the micelles of calcium phosphate stabilized by electrolyte and protein components [50,51]. The nucleus of the micelle consists of $Ca_3(PO_4)_2$, the potential-determining ions are $HPO_4^{2-}$, counter-ions are $Ca^{2+}$, and these are also comprised in the diffused layer. The $Ca_3(PO_4)_2$ micelles are the structural units of the saliva, the mineralization function of which depends on their stability [52]. In the saliva, the micelles are mostly protected from aggregation by the mucin glycoprotein, which, due to its large surface activity, is capable of adsorbing on colloid particles, exhibiting a protective action [53]. The dominant saliva cations (sodium and potassium), together with other ions, determine the osmotic pressure of saliva, its ionic strength, and are comprised in the salt components of the buffer systems. Apparently, comprising the diffused layer, the electrolyte components facilitate the distribution of micelles in the solution volume, not at the interphase, which, in turn, leads to an increase in surface tension due to a decrease in surface active proteins and albumin at the surface. Additionally, a positive correlation of surface tension was found with the total content of seromucoids, which comprise at least 15 various carbohydrate and protein complexes, including prealbumin, alpha-1-acidic glycoprotein, haptoglobin, etc. The high molecular weight proteins of biological fluids have higher surface activity when compared to albumin, which provides for their advantage in the formation of adsorption layers at the air boundary [54]. In this connection, the existence of a negative correlation of the average molecular weight protein toxins and immunoglobulins with surface tension is logical. In addition, the existence of a negative correlation of surface tension was shown with the level of the lipid peroxidation products, which can provide indirect information regarding the lesion degree of the cellular membranes [55,56].

At clusterization of the studied group, depending on the dynamics of surface tension, interesting features were found. The dependence of surface tension on the lifetime of the surface is determined by the irregular nature of the adsorption and desorption processes of surfactants (lipids, proteins, etc.) on the interphase. Thus, at the starting moment of time (t = 0), the surface layer does not contain an excess of surfactant components; therefore, the value of $\gamma_{0.01}$ for most of the biological fluids is close to the surface tension of water and salt solutions, 70–74 mN/m. However, the study results have shown that, cluster 3 is characterized by $\gamma_{0.01}$ value 15.0% higher than the surface tension of water,

while for other clusters this value is below the surface tension of water by 9.7%, 16.6% and 21.7% correspondingly. The diffusion of surfactants to the surface, the rate of overcoming the adsorption barrier (of electrostatic, entropy, or other nature), and processes of restructuring of the adsorbed molecules in the surface layer generally determine the rates of adsorption and surface tension decrease. The values of $\gamma_{1.0}$ (as well as $\gamma_{0.01}$) characterize both the properties of a solvent and adsorption in the short time area, and $\gamma_{max}$ characterizes that in the average surface lifetime area. The presence of low and medium molecular weight surfactants in the biological fluids mostly determine these processes, whereas for high molecular weight fractions (proteins and other compounds), the $\gamma_{max}$ values are determining. Thus, the obtained data has shown that, for cluster 4, which is characterized by the maximal protein level, the highest decrease in $\lambda$ (both $\gamma_{max}$ and $\gamma_{\infty}$) is observed. However, for solutions with high electrolyte content, e.g., cluster 1, the decrease in surface tension is less manifested, which can be explained by the salting-out effect of salt on protein molecules, slowing their adsorption in the interphase [57]. It is also known that chloride ions, which are present in high excess, are capable of adsorbing on the protein surface, causing its recharge [58], which, in its turn, prevents the manifestation of the surface active properties of the latter. Thereby, it should be taken into account that the transition of dissolved compounds from the phase volume to the surface layer is accompanied by desolvation of the adsorbate molecules. The desolvation processes are endothermic and they are characterized by an increase in system entropy, and they significantly affect the thermodynamic characteristics of the adsorption equilibria [59], in particular, they facilitate an increase in surface tension.

In general, the lower the inter-molecular interactions inside the liquid, the lower the surface tension at the liquid-air boundary. It can be hypothesized that, according to the intensity of the inter-molecular interactions, the isolated clusters are arranged, as follows: *cluster 3 > cluster 1 > cluster 2 > cluster 4*. In the selected direction, there is a regular decrease in the area of the crystallization zone (Table 5). Thereby, the maximum value of the Ca/P and Na/K coefficients is characteristic for the cluster 3, which indicates a higher mineralization potential of saliva in this group, and the existence of a more stable colloid system. In addition, the minimum contents of protein and seromucoid (which are apparently involved in the stabilization of the micellar structure, which leads to the growth of surface tension) are noted for this system. A reversed situation is observed for cluster 4: at the minimum value of the Ca/P and Na/K coefficients, the content of protein and seromucoids is maximal, which already determines the minimum surface tension at low drop lifetimes.

It should be noted that $\lambda_0$, which describes a total content of surfactants, is maximal for cluster 3, which may be associated with the presence of high molecular weight compounds of non-protein nature (e.g., lipids and their hydroperoxides) exhibiting their properties at higher drop lifetimes [60]. This explains a drastic decrease of surface tension at t$\to\infty$ for this group.

In general, even after the entire study group was divided into clusters according to surface tension, the crystallization patterns inside the selected groups remain quite heterogeneous. This requires the search for additional factors that can affect the morphology of the resulting crystalline structures. The results of the study showed that the picture of the crystallization of saliva varies, even for the saliva of healthy volunteers. The change in the crystallization patterns is due to a rather wide interval of variation of the biochemical composition of saliva in the norm, which affects the change in surface tension and it leads to a change in the type of crystallization patterns.. However, there are published data on the use for diagnosing both the crystallization of biological fluids and dynamic interfacial tensiometry. In this connection, we would like to focus attention on the fact that the application of these methods in clinical laboratory diagnostics requires an understanding of the nature of variability and the need to take it into account. It should be noted that, varying the nature of crystallization, even within the control group, increases the likelihood of making an incorrect diagnosis when using visual methods to evaluate the crystallization patterns, which significantly limits the use of such diagnostic methods in clinical practice.

## 5. Conclusions

In the course of the study, no statistically significant differences in the saliva crystallization patterns were observed, depending on the age and gender characteristics of the volunteers. A negative correlation of the area of the crystallization zone and the surface tension of saliva is shown. When considering the crystallization patterns, their considerable variability was noted; on this basis, the entire studied sample was divided into four clusters by surface tension. It was established that *cluster 3* most differs in the dynamic characteristics of surface tension; it is characterized by the minimum contents of protein and carbohydrate and protein complexes, as well as by the maximum value of the Ca/P and Na/K coefficients. For this cluster, the maximum area of the crystallization zone and the classical picture of dendritic structures were revealed. Further on, the intensity of intermolecular interactions of the selected clusters are arranged, as follows: *cluster 3 > cluster 1 > cluster 2 > cluster 4*. In this direction, the area of the crystallization zone decreases and the morphological structure of the crystallization patterns changes. For cluster 4, a pronounced intermediate zone appears that distinguishes this group from the rest. However, the revealed heterogeneity of crystallization patterns requires the continuation of research in this direction.

**Author Contributions:** Conceptualization, L.V.B. and E.A.S.; methodology, L.V.B.; investigation, E.A.S. and A.P.S.; data curation, E.A.S.; writing—original draft preparation, L.V.B.; writing—review and editing, A.P.S.; visualization, E.A.S.; supervision, L.V.B. All the authors read and approved the final version of the manuscript.

**Funding:** This research received no external funding.

**Conflicts of Interest:** The authors declare no conflict of interest.

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
