# Peer review of "Morphology of Dried Drop Patterns of Saliva from a Healthy Individual Depending on the Dynamics of Its Surface Tension"

_surfaces, doi:10.3390/surfaces2020029_

Round 1
Reviewer 1 Report
COMMENTS TO THE AUTHOR(S)
The manuscript is well written and well organised. This work allows to establish the relationship between the type of crystallisation patterns and the surface tension of human saliva. This manuscript requires major revisions in order to clarify the main focus of the paper and some aspects of the methodological approach used.
General comment:
In the introduction, the authors discussed the possibility to use the features of crystallisation patterns of biological body fluids in order to identify various pathological processes. They stated that diagnostic methods developed on the basis of the crystallisation of biological fluids use a comparison of the crystallisation patterns of a healthy and sick person. The idea sound interesting. Unfortunately the authors enrolled only healthy volunteers, thus how they can be test the idea to use crystallisation patterns for making diagnosis or monitoring particular diseases?
Specific comments:
Introduction: please include a state of the art regarding the main advantages of using saliva as non-conventional fluid to obtain useful information in the field of therapeutic drug monitoring and for monitoring the health status. For this purpose, the following articles can be useful for the authors:
https://doi.org/10.1371/journal.pone.0028182
https://doi.org/10.1016/j.microc.2017.02.010
https://doi.org/10.1016/j.microc.2017.04.033
Line 39-40, please consider the following articles that describe the variability of the chemical composition:
https://doi.org/10.1016/j.microc.2017.02.032
https://doi.org/10.1371/journal.pone.0114430
Line 51: what does mean “normal conditions”? Maybe their referred to nominally healthy volunteers, if yes please change the words.
Line 61, please include all the information related to the Ethical Committee that authorised the work.
Line 68, I agree with the authors regarding the problem related to the turbidity of saliva, which can significant impact the analysis. Using the procedure proposed by the authors, the supernatant could be not homogeneous making the analysis not reproducible. Thus, the common approach to solve this issue is to filter the sample (e.g. using a syringe filter made of regenerate cellulose with a pore size of 0.2 um), which for sure produce a more homogeneous sample. Thus, did the authors tested this method?
Line 104-115, please include the analytical figures of merit for all the analytical methods.
Line 150-151 (Table 1), please revise all the significant digits, them should be revised according to the reproducibility of the analytical procedures. What is the meaning of Ca/P and Na/K?
Line 178-180 (Table 2), please revise all the significant digits.
Line 183-184 (Table 3), what is the rationale of the subdividing the population using a 10 years range? Maybe the authors should consider to use a frequency distribution histogram. In addition, please revise all the significant digits.
Table 4, please revise all the significant digits.
Line 201-203, are the authors sure that a difference of 3.3% can be distinguished? Again, what is the reproducibility of the analytical methods?
Figure 4, the authors separated the entire population in four clusters, so they obtained a mean and standard deviation or median and interquartile range. Thus, why they did not included the error bars in the graph? What the blue line does mean? It is an example from 4 volunteers? If yes, the error bars should be related to the reproducibility of the analytical procedure.
Line 223-224 (Table 5), please revise all the significant digits.
Line 253-260, please include all the p values. In this way, a reader can evaluate if the correlation among variables are significant or not.
Line 268, the viscosity of saliva is mainly due to the amount of mucins and alpha-amylase. Why the authors did not tested these analytes?
Author Response
General comment:
In the introduction, the authors discussed the possibility to use the features of crystallisation patterns of biological body fluids in order to identify various pathological processes. They stated that diagnostic methods developed on the basis of the crystallisation of biological fluids use a comparison of the crystallisation patterns of a healthy and sick person. The idea sound interesting. Unfortunately the authors enrolled only healthy volunteers, thus how they can be test the idea to use crystallisation patterns for making diagnosis or monitoring particular diseases?
In this paper, we wanted to focus attention on the fact that it is impossible to proceed to the diagnosis with the use of methods that even normally show a large scatter of results. There are literary data on the use of both crystallization of biological fluids and dynamic interfacial tensiometry for diagnostics, but the application of these methods requires an understanding of the nature of variability and the need to take it into account.
Specific comments:
Introduction: please include a state of the art regarding the main advantages of using saliva as non-conventional fluid to obtain useful information in the field of therapeutic drug monitoring and for monitoring the health status. For this purpose, the following articles can be useful for the authors:
https://doi.org/10.1371/journal.pone.0028182
https://doi.org/10.1016/j.microc.2017.02.010
https://doi.org/10.1016/j.microc.2017.04.033
Added the following text: «Use of saliva in the clinical laboratory diagnosis is promising, and non-invasive, simple and inexpensive method for the detection of diseases [13, 14]. Integration of complex techniques, including genomic research, epigenome, transcriptome, proteome, metabolome and microbiome, can detect and quantify the number of biomarkers in saliva [15-18].The study of saliva refers to non-invasive methods and is conducted to assess the age and physiological status, identify somatic diseases, pathology of the salivary glands and tissues of the oral cavity, genetic markers, monitoring of drugs, etc. [19-21].»
Line 39-40, please consider the following articles that describe the variability of the chemical composition:
https://doi.org/10.1016/j.microc.2017.02.032
https://doi.org/10.1371/journal.pone.0114430
Added the following text: «One of the limitations for the widespread use of saliva is the high variability of its normal composition, which requires special attention towards the procedure of sample collection and establishment of reference limits for the content of individual components [28-31].»
Line 51: what does mean “normal conditions”? Maybe their referred to nominally healthy volunteers, if yes please change the words.
The wording has changed.
Line 61, please include all the information related to the Ethical Committee that authorised the work.
Information about the protocol was removed for review, now the changes made to the file.
Line 68, I agree with the authors regarding the problem related to the turbidity of saliva, which can significant impact the analysis. Using the procedure proposed by the authors, the supernatant could be not homogeneous making the analysis not reproducible. Thus, the common approach to solve this issue is to filter the sample (e.g. using a syringe filter made of regenerate cellulose with a pore size of 0.2 um), which for sure produce a more homogeneous sample. Thus, did the authors tested this method?
Conditions of centrifugation are selected in such a way as to completely eliminate the turbidity of saliva. We compared the result with that which can be obtained by filtering. However, we do not filter saliva, because with a large number of samples, which we analyze, filtering takes too much time and this can affect the stability of the detected substances.
Line 104-115, please include the analytical figures of merit for all the analytical methods.
Added the following text: «The pH meter was accurate to ±0.002 pH units and a 3 point calibration was used at pH 4, 7 and 10. The pH of human saliva was measured immediately after collection in triplicate for each participant. Photometric methods were used to determine the concentrations of calcium, phosphorus, magnesium, chlorides, protein, albumin, urea, and seromucoids. As a method of measurement and calculation used the method according to the standard. The resolution of the analyzer allows you to fix the optical density of ±0.0002 units, the accuracy of analyte concentration is determined by the concentration of calibrators (calcium - 2.50 mmol/l, phosphorus - 1.61 mmol/l, magnesium - 0.328 mmol/l, chlorides - 100 mmol/l, protein - 0.50 mg/l, albumin - 40 g/l, urea - 8.33 mmol/l). The error of the method does not exceed 1-2%. Seromucoids concentration is expressed in units of optical density. The concentration of potassium and sodium was calculated according to a previously constructed calibration curve, the error of the method does not exceed 10%. All measurements were carried out in triplicate to ensure reproducible results.»
Line 150-151 (Table 1), please revise all the significant digits, them should be revised according to the reproducibility of the analytical procedures. What is the meaning of Ca/P and Na/K?
Significant numbers corrected. Added the following text: «The ratios Ca/P and Na/K characterize the processes of mineralization and demineralization in the oral cavity [38].»
Line 178-180 (Table 2), please revise all the significant digits.
Significant numbers corrected.
Line 183-184 (Table 3), what is the rationale of the subdividing the population using a 10 years range? Maybe the authors should consider to use a frequency distribution histogram. In addition, please revise all the significant digits.
Significant numbers corrected. Currently, there is no single approach to the formation of age groups in medical research [50]. We have chosen the classification with the smallest step of 10 years, taking into account changes in the nervous, endocrine, bone and muscle systems. In the text of the article added a corresponding link to the literature.
Table 4, please revise all the significant digits.
Significant numbers corrected.
Line 201-203, are the authors sure that a difference of 3.3% can be distinguished? Again, what is the reproducibility of the analytical methods?
We agree with the reviewer's opinion, 3.3% - this is practically the absence of differences, made corrections to the article.
Figure 4, the authors separated the entire population in four clusters, so they obtained a mean and standard deviation or median and interquartile range. Thus, why they did not included the error bars in the graph? What the blue line does mean? It is an example from 4 volunteers? If yes, the error bars should be related to the reproducibility of the analytical procedure.
Graph 4 shows just examples of curves corresponding to each cluster; these are not averaged curves, therefore there are no error columns.
Line 223-224 (Table 5), please revise all the significant digits.
Significant numbers corrected.
Line 253-260, please include all the p values. In this way, a reader can evaluate if the correlation among variables are significant or not.
Thus, a positive correlation to pH was found for the cluster 1 (r=0.5628; p=0.0042) and negative one to the protein and albumin contents (r=-0.2966; p=0.0426 and r=-0.5785; p=0.0079). For the cluster 2, a positive relationship of surface tension with potassium and chloride levels was established (r=0.3238; p=0.0446 and r=0.5388; p=0.0092). For the cluster 3, a negative correlation with the Ca/P ratio was shown (r=-0.5952; p=0.0051), as well as with protein and albumin levels (r=-0.7857; p=0.0002 and r=-0.5270; p=0.0027), and a positive relationship with the Na/K ratio (r=0.3571; p=0.0063).
Line 268, the viscosity of saliva is mainly due to the amount of mucins and alpha-amylase. Why the authors did not tested these analytes?
We determined the content of individual fractions of glycoproteins, which include mucin, in particular seromucoids and sialic acids, however, statistically significant differences were found only for the seromucoid fraction, which is listed in table 6. For amylase, there were no differences between the clusters, however, the content of mucin and amylase is summarized in total protein. For protein patterns identified and described in the article. However, in our opinion, precisely the differences in the content of electrolyte components can make a greater contribution to the dynamics of surface tension and changes in the microcrystallization patterns of saliva.
Reviewer 2 Report
Dear Authors
The idea of the project looks fabulous. I went through with whole manuscripts, research idea, protocols all are very well defined. Looks very interesting scientific story and I enjoyed.
I just see one lacking is the outcome and future direction on this topic. Please elaborate more on the saliva diagnosis.
There are a few recommended papers for your help. Read them and cite appropriately in the Introduction and discussion headings.
a) Sahibzada, Haafsa Arshad, et al. "Salivary IL-8, IL-6 and TNF-α as potential diagnostic biomarkers for oral cancer." Diagnostics 7.2 (2017): 21.
b) Khan, R., Khurshid, Z., & Yahya Ibrahim Asiri, F. (2017). Advancing point-of-care (PoC) testing using human saliva as liquid biopsy. Diagnostics, 7(3), 39.
Author Response
There are a few recommended papers for your help. Read them and cite appropriately in the Introduction and discussion headings.
a) Sahibzada, Haafsa Arshad, et al. "Salivary IL-8, IL-6 and TNF-α as potential diagnostic biomarkers for oral cancer." Diagnostics 7.2 (2017): 21.
b) Khan, R., Khurshid, Z., & Yahya Ibrahim Asiri, F. (2017). Advancing point-of-care (PoC) testing using human saliva as liquid biopsy. Diagnostics, 7(3), 39.
Added the following text in the Introduction: «Use of saliva in the clinical laboratory diagnosis is promising, and non-invasive, simple and inexpensive method for the detection of diseases [13, 14]. Integration of complex techniques, including genomic research, epigenome, transcriptome, proteome, metabolome and microbiome, can detect and quantify the number of biomarkers in saliva [15-18].The study of saliva refers to non-invasive methods and is conducted to assess the age and physiological status, identify somatic diseases, pathology of the salivary glands and tissues of the oral cavity, genetic markers, monitoring of drugs, etc. [19-21].»
Added the following text in the Discussion:«Thus, the results of the study showed that the picture of the crystallization of saliva varies even for the saliva of healthy volunteers. The change in the crystallization patterns is due to a rather wide interval of variation of the biochemical composition of saliva in the norm, which affects the change in surface tension and leads to a change in the type of crystallograms. However, literature data on the use of both the crystallization of biological fluids and dynamic interfacial tensiometry for diagnosis is. In this connection, we would like to focus attention on the fact that the application of these methods in clinical laboratory diagnostics requires an understanding of the nature of variability and the need to take it into account.»
Round 2
Reviewer 1 Report
Dear authors, the revised version of the paper is suitable for the publication in Surfaces Journal.
Regards